# CL-20-Based Cocrystal Energetic Materials: Simulation, Preparation and Performance

**DOI:** 10.3390/molecules25184311

**Published:** 2020-09-20

**Authors:** Wei-qiang Pang, Ke Wang, Wei Zhang, Luigi T. De Luca, Xue-zhong Fan, Jun-qiang Li

**Affiliations:** 1Xi’an Modern Chemistry Research Institute, Xi’an 710065, China; zhuzhangmangxiewk@163.com (K.W.); xuezhongfan@126.com (X.-z.F.); llijq@sohu.com (J.-q.L.); 2Science and Technology on Combustion and Explosion Laboratory, Xi’an 710065, China; 3School of Chemical Engineering, Nanjing University of Science and Technology, Nanjing 210094, China; wzhang@njust.edu.cn; 4Department of Aerospace Science and Technology, Politecnico di Milano, 20156 Milan, Italy; luigi.t.deluca@gmail.com

**Keywords:** molecular dynamic simulation, CL-20, cocrystal energetic materials, preparation, characterization

## Abstract

The cocrystallization of high-energy explosives has attracted great interests since it can alleviate to a certain extent the power-safety contradiction. 2,4,6,8,10,12-hexanitro-2,4,6,8,10,12-hexaaza-isowurtzitane (CL-20), one of the most powerful explosives, has attracted much attention for researchers worldwide. However, the disadvantage of CL-20 has increased sensitivity to mechanical stimuli and cocrystallization of CL-20 with other compounds may provide a way to decrease its sensitivity. The intermolecular interaction of five types of CL-20-based cocrystal (CL-20/TNT, CL-20/HMX, CL-20/FOX-7, CL-20/TKX-50 and CL-20/DNB) by using molecular dynamic simulation was reviewed. The preparation methods and thermal decomposition properties of CL-20-based cocrystal are emphatically analyzed. Special emphasis is focused on the improved mechanical performances of CL-20-based cocrystal, which are compared with those of CL-20. The existing problems and challenges for the future work on CL-20-based cocrystal are discussed.

## 1. Introduction

Energetic materials (EMs) are widely used in military and civilian applications, such as weaponry, aerospace explorations and fireworks. However, both the power and safety of energetic materials are the most concern in the field of energetic materials application, but there is an essential contradiction between them: the highly energetic materials are often not safe, and, at present, the rareness of pure low-sensitive and highly energetic explosive has been found [1,2,3]. For a long time, in order to obtain EMs with lower sensitivity, the modifications of existing explosives have often focused mainly on recrystallizing with solution and coating with polymer [4,5,6]. However, these traditional methods cannot markedly reduce the sensitivities of existing explosives by only modifying morphology or diluting power, due to the unchanging inherent structures of explosive molecules [7,8]. Due to the stringent requirements for both low-sensitivity and high-power simultaneously, the cocrystallization of explosive, a technique by which a multi-component crystal of several neutral explosive molecules forms in a defined ratio through non-covalent interactions (e.g., H-bond, electrostatic interaction, etc.) [9,10], has attracted great interest since it can alleviate, to a certain extent, the power-safety contradiction [11,12]. The new energetic cocrystals can potentially exhibit decreased sensitivity, higher performance through increased packing efficiency or oxygen balance, or improve aging due to altered bonds to specific groups on constituent molecules [13,14,15]. Recently a lot of cocrystal explosives have been synthesized and characterized [16,17,18,19,20]. An evaluation of the power and safety of energetic cocrystals has been carried out, and cocrystallization is becoming increasingly hot in the field of energetic materials [21]. It was found that the stability (sensitivity) and detonation performance (energy, detonation velocity, and detonation pressure, etc.) of cocrystal explosive could be influenced by the molar ratio of molecular combination. Generally, when a cocrystal has too much content of high-energy explosives, the packing density and detonation performance will be increased, with possible increase in explosive sensitivity. On the contrary, the sensitivity will be decreased in a cocrystal explosive with a high content of low-energetic or non-energetic explosives. The searches for stable insensitive and highly-energetic explosive is the primary goal in the field of energetic material chemistry. Therefore, the molar ratios of two or more kinds of explosive components should be controlled in a reasonable scope, and it is very necessary to clarify the influence of the ratio of molecular combination on the stability and detonation performance of cocrystal explosives, such as packing density, oxygen balance, detonation velocity, and detonation pressure, etc. [22].

The compound 2,4,6,8,10,12-hexanitro-2,4,6,8,10,12-hexaazaisowurtzitane (CL-20) is currently known to be one of the most powerful explosives [23], and it is also a superior alternative to HMX for applications in low signature rocket propellants [24,25,26] and gun propellants [27]. However, the disadvantage of CL-20 is its increased sensitivity to mechanical stimuli and cocrystallization of CL-20 with other compounds, which may provide a way to decrease its sensitivity [28]. Moreover, the relationship of the reaction kinetic process with temperatures and densities for pyrolysis of CL-20/TNT cocrystal was investigated using a reactive force field (ReaxFF) molecular dynamics simulation. The evolution distribution of potential energy and total species, decay kinetics and kinetic parameters for thermal decomposition reaction of CL-20 and TNT were analyzed. It was shown that the breaking of −NO_2_ bond from CL-20 molecules is the initial reaction pathway for the thermal decomposition of the cocrystal. With increasing the cocrystal density, the reaction energy barrier of CL-20 and TNT molecule decomposition increases correspondingly. The decomposition process of TNT has an inhibition action on the decomposition of CL-20 [29]. Meanwhile, the combustion behavior, flame structure, and thermal decomposition of bi-molecular crystals of CL-20 with glycerol triacetate (GTA), tris [1,2,5]oxadiazolo [3,4-b:3′,4′-d:3′’,4′’-f]azepine-7-amine (ATFAz), 4,4′’-dinitro-ter-furazan (BNTF), oxepino [2,3-c:4,5-c’:6,7-c’’] trisfurazan (OTF), oxepino [2,3-c:4,5-c’:6,7-c’’] trisfurazan-1-oxide (OTFO) were studied [30]. It was found that the introduction of volatile and thermally stable compounds into the composition with CL-20 decreased the thermal stability of CL-20. The combustion mechanism of the CL-20 cocrystal depends both on the burning rate of the second component and its volatility. While there are few overview papers on CL-20-based cocrystal’s simulation, preparation and characterization. Thus, in this paper, the achievements in the intermolecular interaction of five types of CL-20-based cocrystals (CL-20/TNT, CL-20/HMX, CL-20/FOX-7, CL-20/TKX-50 and CL-20/DNB) by using molecular dynamic simulation are reviewed. The preparation and performance of CL-20-based cocrystals are emphatically analyzed, and the existing problems and challenges in the future work are reviewed. The aim of this work is mainly to explore the nature of the formation of the different CL-20-based cocrystal morphology and clarify the influence of the ratio of molecular combination on the stability and detonation performance of CL-20-based cocrystal explosives.

## 2. Modelling and Simulation

### 2.1. Molecular Structures of Cocrystal Monomer

In order to simulate the properties of cocrystals, the molecular structures of each material can be constructed. Table 1 lists the molecular structures of different energetic materials formed cocrystals.

Simulation details: Andersen was set as the temperature control method (298.15 K). COMPASS force-field was assigned. Summation methods for electrostatic and van der Waals were Ewald and Atom-based, respectively. The accuracy for the Ewald method was 1.0 × 10^−4^ kcal·mol^−1^. Cutoff distance and buffer width for the Atom-based method were 15.5 Å and 2.0 Å, respectively; 1.0 f. of time step was set for MD processes, and the total dynamic time was performed with 100,000 fs. All the MD calculations were carried out with MS 7.0 [31].

Because the binding energies of cocrystals obtained from the MD calculations are of different numbers of molecules, i.e., different supercells and different molecular molar ratios, those of the same number of molecules should be adopted as a standard for assessing the component interaction strengths and stabilities of cocrystals. Therefore, according to our recent investigation [32], an energy correction formula for binding energy was used to standardize (uniform) the differences caused by diverse supercells and different molecular molar ratios as follows:*E*_b_* = *E*_b_·*N*_0_/*N*_i_,(1)
where *E*_b_^*^ denotes the binding energy after corrected, and *N*_i_ and *N*_0_ are the number of molecules for different supercells and a standard pattern (molar ratio in 1:1), respectively. The binding energy *E*_b_ is calculated by the following formula:*E*_b_ = *E*_tot_ − (n*E*_CL-20_ + m*E*_CM_),(2)
where *E*_tot_, *E*_CL-20_ or *E*_CM_ is single point energy of cocrystal or monomer; *n* and m are the number of monomers in cocrystal.

### 2.2. CL-20/TNT System

CL-20 is a complex cage compound, which can be regarded as a six-membered ring and two five-membered rings arranged together, and it has six nitro groups. The preparation of CL-20/TNT cocrystal by the solvent method has attracted much attention worldwide due to its high-energy, insensitive and cheap explosives. The density and melting point of the CL-20/TNT cocrystal explosive (1.91 cm, 136 °C) are between CL-20 (2.04 g/cm^3^, 210 °C) and TNT (1.70 g/cm^3^, 81 °C), and the sensitivity is nearly double reduced that of CL-20 [11]. Figure 1 shows the intermolecular hydrogen bond of CL-20/TNT cocrystal.

The non-bond distances of O (3), H (5) in CL-20 and H (8), O (13) in TNT are 0.231 nm and 0.244 nm, respectively, which are smaller than the sum of their van der Waals radii (0.272 nm [33]), indicating the existence of intermolecular hydrogen bonds. Through the interaction of C-H…O hydrogen bond between the intermolecular, CL-20 and TNT molecules are connected as the form of “zigzag chain” structure, and combined to form the stable structure of CL-20/TNT cocrystal, indicating that the intermolecular hydrogen bond makes the arrangement of CL-20 and TNT molecules in the crystal more regular, and the molecular packing is close with higher crystal density (1.84 g·cm^−3^).

Meanwhile, the cohesive energy density (CED) of CL-20/TNT cocrystal and its CL-20/TNT composite mixture at different temperatures were investigated [11], results show in Table 2 and Figure 2. With the increase of temperature, CED and its component VDW force and electrostatic force decrease gradually, which is negatively related to the law that the sensitivity increases gradually with the increase of temperature. Under certain conditions, CED, which reflects the energy required for phase transition, can be used to determine the relative thermal sensitivity.

### 2.3. CL-20/HMX System

For HMX, if CL-20 and HMX can form the cocrystal in a certain proportion, the sensitivity of CL-20 can be significantly reduced on the basis of its energy reduce little; on the other hand, the cost of explosive can be significantly reduced and its application range can be expanded [32].

The properties of hexanitrohexaazaisowurtzitane (CL-20)/cyclotetramethylenete-tranitramine (HMX) cocrystal were simulated and compared with those of CL-20/HMX mixture with the molar ratio of CL-20 and HMX as 2:1 [2]. Simulation and calculation results show that the cocrystal process of CL-20/HMX can significantly improve the anti-deformation ability and ductility of the system, and the tensile modulus of the cocrystal structure is greater than that of the mixture. The maximum bond length (*L*_max_) decreases in the order of: [CL-20/HMX mixture] > [ε-CL-20] > [β-HMX] > [CL-20/HMX cocrystal]. The structure of CL-20/HMX mixture is sensitized by the predominant interaction of Van der Waals force (Table 3), showing that the two components in the mixture system can be stably adsorbed and have good physical compatibility. The cohesive energy density (CED) value of CL-20/HMX cocrystal structure is far greater than that of CL-20/HMX mixture, and with the increase of temperature, the cohesive energy density of the two components decreases and the structural stability becomes worse (Table 4). The low sensitivity of CL-20/HMX cocrystal system is caused by the existence of hydrogen bond CH–O with relatively short length.

### 2.4. CL-20/FOX-7 System

For CL-20/FOX-7 cocrystal, the unit cell models of CL-20 and FOX-7 were constructed according to their experimental cell parameters, respectively. Initial models were obtained by discovering module in the COMPASS force field; 1.0 × 10^−5^ kcal mol^−1^ of accuracy was required. The CL-20 and FOX-7 crystal morphologies in vacuum were predicted by the Growth morphology model. Different cocrystal molar ratios can be treated by the substituted method: molecules of CL-20 supercells were substituted by an equal number of FOX-7 at molar ratios of 8:1, 5:1, 3:1, 2:1, and 1:1 (CL-20/FOX-7). Molecules of FOX-7 supercells were substituted by an equal number of CL-20 (ε-, γ- and β-forms) at molar ratios of 1:2, 1:3, 1:5, and 1:8 (CL-20/FOX-7). Substituted molecules in this method were determined by the Miller indices hkl. For the substituted models, NVT ensembles were selected.

Five growth faces (0 1 1), (1 01), (1 0 -1), (0 0 2), and (1 1 -1) and random face of ε-, γ- and β-CL-20 were selected to study the binding energies of the cocrystals with FOX-7 in different molar ratios. These growth faces and a random face of FOX-7were selected to study the binding energies of cocrystals. Based on the MD simulation of substituted models, *E*_b_^*^ of the ε-, γ-, and β-CL-20 cocrystal explosives with FOX-7 on the different cocrystal faces in the different molar ratios are calculated (Table 5). As can be seen, except for the random trend, there is a trend that the strongest binding energies *E*_b_^*^ in the γ-CL-20/FOX-7 are larger than those in ε-CL-20/FOX-7 and β-CL-20/FOX-7. Furthermore, for γ-CL-20/FOX-7, the binding energies *E*_b_^*^ on the (1 1 0) and (1 0 -1) cocrystal faces of FOX-7 in 1:2 and those on the (1 0 -1) and (1 1 0) faces of γ-CL-20 in 1:1 are larger than the other cases. These results indicate that FOX-7 may prefer cocrystalizing with γ-CL-20 on the (1 1 0) and (1 0 -1) faces of FOX-7 in 1:2, or on the (1 0 -1) and (1 1 0) faces of γ-CL-20 in 1:1. It is noted that, as mentioned above, for the cocrystal with the excess ratio of FOX-7, the FOX-7 supercells were substituted by an equal number of CL-20, while in the cocrystal with the excess ratio of CL-20, the CL-20 supercells were substituted by FOX-7.

### 2.5. CL-20/TKX-50 System

TKX-50 is an ionic salt structure with two hydroxylamine cations, and it can easily form hydrogen bonds between H of -NH_3_^+^ in TKX-50 and O of –NO_2_ in CL-20, which is in good agreement with the results of surface electrostatic potential energy analysis. The surface electrostatic potential energy analysis of CL-20 and TKX-50 is useful for exploring the formation mechanism of CL-20/TKX-50 cocrystal. CL-20/TKX-50 cocrystals with the mole ratio of 1:1, 1:2, 1:3 and 2:3 were prepared by means of the solvent–nonsolvent method [34]. Compared with the raw materials and the mixture, the XRD spectra of CL-20/TKX-50 cocrystals have no obvious change when the feed ratio is 1:1, 1:3 and 2:3, and it can be considered that the eutectic of CL-20/TKX-50 cocrystals has not been successfully prepared. Finally, the optimal mole ratio is 1:2. Figure 3 shows the surface electrostatic potential energy distribution map of CL-20 (a) and TKX-50 (b), and the equilibrium structure of CL-20/TKX-50 cocrystal model (c), a schematic diagram of intermolecular interaction of CL-20 and TKX-50 model (d).

### 2.6. CL-20/DNB System

As a cheap and insensitive explosive, DNB is often used as an alternative explosive for TNT, but its energy is not ideal. If CL-20 and DNB can be combined in the same lattice through noncovalent bond at the molecular level through cocrystal technology to form an explosive crystal with unique structure, it is expected that the sensitivity and cost of CL-20 will be greatly reduced without significant energy reduction, so as to expand the application range of CL-20 [12]. In order to improve the hazardous performance of CL-20, 1,3-dinitrobenze (DNB) was introduced to CL-20 to form CL-20/DNB cocrystal with 1:1 mole ratio. The primitive cell of CL-20/DNB cocrastal was shown in Figure 4.

The molecular dynamics simulation of 2,4,6,8,10,12-hexanitrohexaazaisowurtzitane (CL-20)/1,3- dinitrobenzene (DNB) cocrystal and CL-20/DNB cocrystal with two polymer binders, hydroxyl-terminated polybutatiene (HTPB) and polyethylene glycol (PEG) respectively were conducted [3]. Results indicate that the binding energy of CL-20/DNB/PEG is larger than that of CL-20/DNB/HTPB, predicting that the stability and compatibility of the former is better than those of the latter (Table 6). In comparison with CL-20/DNB cocrystal, an addition of a small amount of binder (HTPB or PEG) decreases the elastic constants (*C*_ij_), tensile modulus (*E*), bulk (*K*) and shear modulus (*G*), while Cauchy pressure (*C*_12_–*C*_44_) and *K*/*G* value increase, showing that the stiffness of the polymerbonded explosives (PBXs) system is weaker, and its ductibility is better.

In addition, the initial thermal decomposition pathways, as well as some important products generating mechanism of CL-20/DNB cocrystal at high temperatures (2000, 2500 K and 3000 K), were studied by reactive molecular dynamics simulations using ReaxFF force field [4]. Results show that with the increasing of temperature during the thermal decomposition process, the time to balance and potential energy decrease, while the quantity of products increases. All the CL-20 molecules decompose faster than that of DNB, and as the temperature rises, the decomposition rate of DNB increases significantly. According to the product identification analysis, the main thermal decomposition products are NO_2_, NO, N_2_, H_2_O, HNO_3_, HON, HONO and CO_2_ for the cocrystal. The major initial decomposition mechanism is the breaking of N−NO_2_ in the CL-20 and C−NO_2_ in the DNB, which contributes to the formation of NO_2_. Then, the number of NO_2_ increases to the peak rapidly and decreases subsequently (Figure 5). After the NO_2_→ONO rearrangement, it participates in other reactions and eventually N_2_, NO, HONO, HON, H_2_O occur, and so on. In addition, the simulation results indicate that carbon-containing clusters formed in the later stage of decomposition at 2500 K and 3000 K, which is a common phenomenon during the detonation of rich carbon-containing explosives.

Meanwhile, the effect of CL-20/DNB cocrystallizing and mixing on the sensitivity, binding energy, mechanical properties and thermal decomposition on the molecular level were investigated under the condition of the COMPASS force field. Results indicate that the cocrystallizing and mixing can reduce the sensitivity of CL-20, increase that of DNB, and the cocrystallizing effect is more obvious and stable [35].

## 3. Preparation

For CL-20/TNT cocrystal, the CL-20/TNT cocrystal with 1:1 mole ratio was prepared by means of the recrystallization method at room temperature, and with ethyl acetate as solvent [8]. It was found that the morphology of CL-20/TNT cocrystal is different from that of CL-20 and TNT. The crystal of CL-20 is “spindle” shape and TNT is an irregular block crystal. However, CL-20/TNT cocrystal is a prismatic crystal with a smooth and complete surface and uniform size (Figure 6). The average particle size is 270 μm, indicating that the designing and preparation of cocrystal can effectively change the shape and size of explosive cocrystals.

At the same time, ultrafine CL-20/TNT cocrystal explosive was prepared by a spray-drying method [20]. Results show that the prepared samples are not the mix of CL-20 and TNT but rather ultrafine CL-20/TNT cocrystal explosives. The particle sizes of the cocrystal explosives were lower than 1 μm and they can aggregate into many microparticles, which are spherical in shape and 1–10 μm in size (Figure 7). The thermal decomposition process can be divided into two stages. The peak temperatures of exothermic decomposition for the first and second stages are 218.98 °C and 253.15 °C, respectively (Figure 8). The characteristic drop height of CL-20/TNT cocrystal explosives is 49.3 cm, which increases by 36.2 cm compared with that of raw CL-20 (Table 7).

The ultrafine CL-20/HMX cocrystal explosive was prepared by an ultra-highly efficient mixing method [6]. The prepared samples were regular block-like ultrafine CL-20/HMX cocrystal explosives with a uniform particle size of less than 1 μm (Figure 9), which appeared new stronger diffraction peaks at 11.558°, 13.264°, 18.601°, 24.474°, 33.785°, 36.269°. The purity of the CL-20/HMX cocrystal explosive was 92.6%.

Moreover, the micro/nano CL-20/HMX energetic cocrystal materials were prepared by mechanical ball milling, and characterized using SEM [7]. Results show that after a milling time of 120 min, CL-20/HMX cocrystals prepared under optimal conditions are spherical in shape and the particle size is 80–250 nm. Compared with respective energetic monomers, the prepared micro/nano CL-20/HMX energetic cocrystal materials exhibit unique crystal structure and thermal decomposition properties (Figure 10).

In addition, nano-sized CL-20/HMX cocrystals with a mean particle size of 81.6 nm were prepared under the conditions of the 0.3 mm diameter of milling balls [9]. The impact sensitivity was conducted to evaluate the impact safety performance of the CL-20/HMX cocrystal (Table 7). The micro-morphology of the explosives is near-spherical and the particle size reveals a normal distribution (Figure 11). Before and after milling, the element composition and molecular structure of CL-20/HMX did not change in comparison to the raw CL-20 and HMX, while there are new crystal phases in the formed cocrystal. The characteristic drop height (*H*_50_) of CL-20/HMX was 32.62 cm with 5 kg hammer, which is lower than that of raw materials and mixture, indicating that the mechanical ball milling method is not simply to mix the two explosives physically, but to form the cocrystal between the explosive molecules through the hydrogen bond. Meanwhile, the crystal technology can improve the impact safety of explosives.

Nano-scale CL-20/HMX cocrystal of CL-20 and HMX in 2:1 molar ratio with a mean size below 200 nm was prepared by bead milling [36]. Figure 12 shows the morphological evolution of the crystal particles. It was found that most of the coformer crystal particles are ~1 μm (Figure 12a). The mean particle size of the discrete coformers substantially decreased after 10 min of milling (Figure 12b), with no conversion to the cocrystalline material. Plate-like crystal particles with dimensions less than 500 nm started to appear in the specimen being milled for 20 min, as indicated by the arrow in Figure 12c. The plate-like particles were assigned to the 2CL-20·HMX cocrystal as (1) the 2CL-20·HMX cocrystal is known to have a plate-like morphology; (2) the appearance of these particles and the diffraction peaks of the 2CL-20·HMX cocrystal in the XRD pattern occurred at the same time; and (3) more plate-like particles were observed in the specimens upon further milling (Figure 12d and e, respectively). The observation of these relatively large cocrystal particles seems to be contradicting to the intensive collisions between the grinding media and particles and between the particles themselves occurring during the milling process. It is possible that some growth occurs during the drying of the sampled specimens.

Preparation procedure: 438.0 mg (1 mmol) CL-20 and 472.3 mg (2 mmol) TKX-50 were dried in a beaker at 40 °C for 3 h, 30 mL DMF was added to the beaker, heat to 80 °C in the water bath, and ultrasonic for 30 min until the drug is completely dissolved. Then, 100 mL chloroform was put into the flask. The completely dissolved solution was dropped into the flask at the rate of 0.8 mL/min, and the magnetic stirring speed was kept at 1000 r/min. After dropping, keep the original speed and continue stirring for 1 h. After standing for 3 h and filtering, the filtered samples were dried in a vacuum for 3 h to obtain CL-20/TKX-50 ultrafine cocrystal samples. The performance was compared with that of CL-20/TKX-50 mechanical mixture in the same mole ratio.

To decrease the sensitivity of CL-20, the CL-20/TKX-50 cocrystal explosive was prepared by solvent-nonsolvent method. The surface electrostatic potentials of CL-20 and TKX-50 were analyzed and the possible non-covalent bonding between cocrystal molecules was predicted [34]. Results shown that the prepared CL-20/TKX-50 cocrystal has a flat sheet shape, the formation, disappearance, shift and change of intensity of peaks been proved the formation of a new lattice structure. The crystal morphology of CL-20 and TKX-50 is nearly spherical shape, and the particle size distribution is relatively uniform, and the particle size is 1 μm. However, the morphology of CL-20/TKX-50 cocrystal shows a long and thin lamellar structure, which is quite different from that of CL-20 and TKX-50, and the particle size of CL-20/TKX-50 cocrystal is 10 μm, indicating that the cocrystal process can not only change the morphology of the original crystal, but also prove the formation of a new crystal (Figure 13).

## 4. Energetic Performance

In order to study the thermal decomposition properties of CL-20/TNT cocryatal explosives, the samples of CL-20/TNT cocryatals were tested by DSC (TA Instruments, New Castle, DE, USA) at the heating rate of 10 °C·min^−1^, the curve shown in Figure 14. As is shown there are three stages of the thermal decomposition process of CL-20/TNT cocrystal, the maximum endothermic peak temperature is 143.5 °C, which is higher than that of the melting point of TNT (81 °C [37]). With the increase of temperature, the hydrogen bond between the cocrystal molecules breaks and the molecular structure is destroyed [38]. There are two processes of exothermic decomposition at 222.6 °C and 250.1 °C, respectively. Compared with the maximum exothermic peak value of CL-20 (321.5 °C [33]) and TNT (245 °C [39]), the exothermic peak of CL-20/TNT cocrystal shifts, indicating that the cocrystal changes the thermal decomposition characteristics of the raw materials and endows the cocrystal with unique thermal decomposition behavior.

The impact sensitivity value of CL-20/TNT cocrystal is *H*_50_ = 28 cm. Compared with CL-20 (*H*_50_ = 15 cm), the impact sensitivity of CL-20 is significantly reduced by 87%. Low-sensitivity TNT and high-sensitivity CL-20 combine to form a cocrystal at the molecular scale through eutectic technology, which changes the internal composition and crystal structure of explosives compared with traditional methods. In addition, due to the intermolecular hydrogen bond, on the one hand, it increases the stability of the molecular system of the cocrystal, on the other hand, it improves the anti-vibration of the cocrystal molecule to the mechanical external force, so the desensitization effect is obvious. Through the eutectic technology, it can effectively realize the desensitization of high sensitive explosive and improve its safety performance.

The crystal morphology, particle size and sensitivity of prepared CL-20/HMX cocrystal were characterized [6]. The thermal decomposition process of cocrystal explosives had only one exothermic decomposition stage with peak temperatures of 248.3 °C. The enthalpy for the exothermic decomposition of the cocrystal (2912.1 J·g^−1^) was remarkably higher than that of the physical mixture of CL-20 and HMX (1327.3 J·g^−1^) (Figure 15). The friction sensitivity of CL-20/HMX cocrystal explosive was 84%, which was decreased by 16% compared with original CL-20, and the characteristic height of the cocrystal was increased by 28.6 cm and 11.5 cm compared with original CL-20 and HMX, respectively (Table 8). The compatibility of CL-20/HMX cocrystal with components of solid propellant shown that the prepared CL-20/HMX cocrystal was compatible with NG/BTTN, AP and Al powder, while incompatible with HGAP, N-100.

At the same time, the DSC curves of CL-20/HMX cocrystal were prepared by means of mechanical ball milling, and compared with the CL-20/HMX mixture [7]. The mechanical sensitivity of the micro/nano CL-20/HMX energetic cocrystal materials was reduced obviously compared to that of raw HMX, while the energy output property was equivalent to that of raw CL-20.

In order to compare the effect of the prepared CL-20/TKX-50 cocrystal on the thermal decomposition of each component, the DSC curves of CL-20, TKX-50, CL-20/TKX-50 mixture and CL-20/TKX-50 cocrystal were investigated [34] (Figure 16). The exothermic decomposition peaks of CL-20 and TKX-50 were 240.2 and 234.8 °C, respectively, and there were two exothermic decomposition peaks in the decomposition process of the CL-20/TKX-50 mixture, indicating that the decomposition process of the CL-20/TKX-50 mixture is obviously a simple superposition of CL-20 and TKX-50. The first decomposition peak appears at 229.8 °C, corresponding to the exothermic decomposition of part of TKX-50; the second peak appears at 242.5 °C, corresponding to the exothermic decomposition of CL-20. For the CL-20/TKX-50 cocrystal decomposition, with the increase of temperature, the first exothermic decomposition peak of CL-20/TKX-50 cocrystal appears at 171.6 °C, which can be attributed to the destruction of hydrogen in the cocrystal structure and the decomposition of a small amount of TKX-50. Subsequently, a large number of cocrystal products begin to decompose, and the second exothermic decomposition peak appeared at 222.8 °C, indicating the formation of hydrogen bond and the existence of a new structure between CL-20/TKX-50 cocrystal molecules.

In addition, the DSC curves of CL-20, DNB, CL-20/DNB cocrystal were shown in Figure 17. There are three (one is endothermic melting and two exothermic decompositions) thermal decomposition stages. The melting temperature of cocrystal is 136.4 °C, 44.7 °C higher than that of raw material DNB (91.7 °C) in the melting stage, indicating that the cocrystal decomposes into liquid DNB at this point. There are two exothermic peaks at 216.9 °C and 242.9 °C in the exothermic decomposition stage, which was attributed to the exothermic decomposition of two single components after cocrystal decomposition. Moreover, the crystal density of CL-20/DNB is higher than that of CL-20/TNT, and the sensitivity of DNB is lower than that of TNT. Thus, the sensitivity of CL-20/DNB is lower than that of CL-20/TNT. In addition, the cost of the DNB is significantly lower than that of TNT. Therefore, the cocrystal may become an excellent explosive with high-energy, insensitive and low-cost prospective ingredients in explosives and propellants.

The impact sensitivity, the calculated detonation of the CL-20, TKX-50, CL-20/TKX-50 mixture and CL-20/TKX-50 cocrystal are shown in Table 8. The characteristic drop height of CL-20/TKX-50 cocrystal is lower than that of TKX-50, but significantly higher than that of CL-20, β-HMX and CL-20/TKX-50 mixture, indicating that the impact sensitivity of CL-20/TKX-50 cocrystal is much lower. The detonation velocity and detonation pressure of CL-20/TKX-50 cocrystal are slightly lower than that of CL-20, but the detonation performance is obviously improved compared with β-HMX, indicating CL-20/TKX-50 cocrystal has good detonation performance.

## 5. Existing Problems and Challenges

As we all know, it is of great significance for the preparation of energetic materials with high-energy and low-sensitivity to use eutectic technology to modify the single high-energy explosives. While the development of an energetic cocrystal is still in the exploratory stage, the main challenges for cocrystal at present are as follows: (1) the optimal preparation method of energetic cocrystal still needs to be simplified and simulated. The solvent evaporation method is mainly used to explore the preparation conditions of cocrystal by experience at present and many experiments with different stoichiometric ratios cannot meet the engineering requirements. (2) The effective cocrystal formation principle and formation mechanism of energetic compounds under the guidance of thermodynamics are not fully revealed, which greatly hinders the controllable construction of energetic cocrystal compounds [40]. The main challenges for us in the future are possibly: (1) the design and formation mechanism of energetic cocrystal can be draw lessons from the other research fields, the cocrystal is formed by self-assembly of hydrogen bonding and π–π stacking interactions; (2) on the basis of the existing mature eutectic technology, a safe, efficient and practical synthesis method suitable for energetic cocrystal should be improved, promoting the theoretical research of cocrystal formation, and simulate and design the formation of energetic cocrystal from the theoretical level [41].

## Figures and Tables

**Figure 1 molecules-25-04311-f001:**
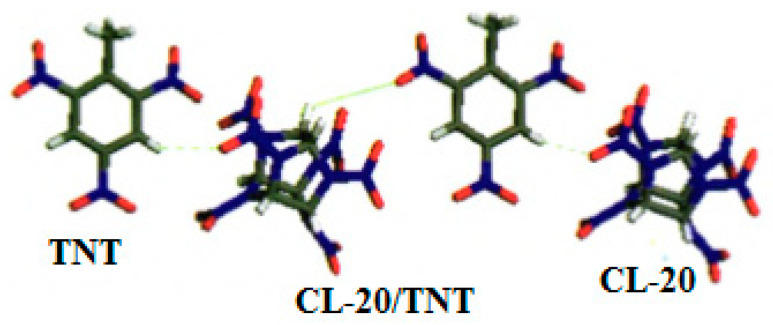
Intermolecular hydrogen bond of CL-20/TNT cocrystal [8], with permission from Han Neng Cai Liao, 2012.

**Figure 2 molecules-25-04311-f002:**
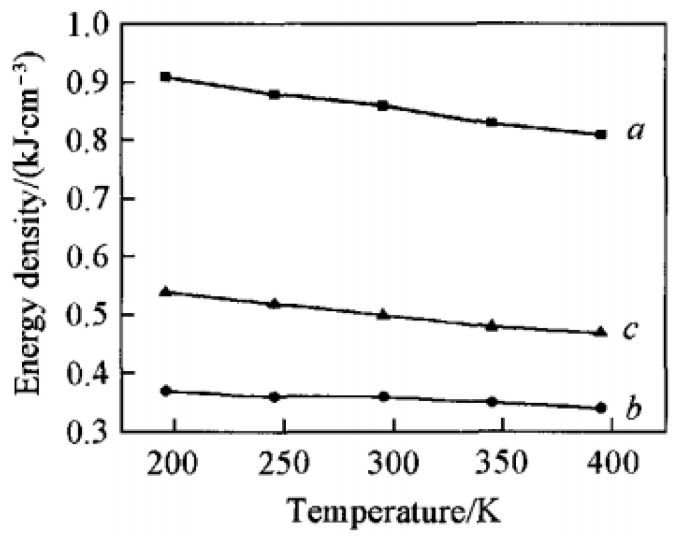
CED (a), vdw (b) andelectrostatic (c) energies of CL-20/TNT cocrystal vs. temperatures [11], with permission from Chemical Journal of Chinese Universities, 2016.

**Figure 3 molecules-25-04311-f003:**
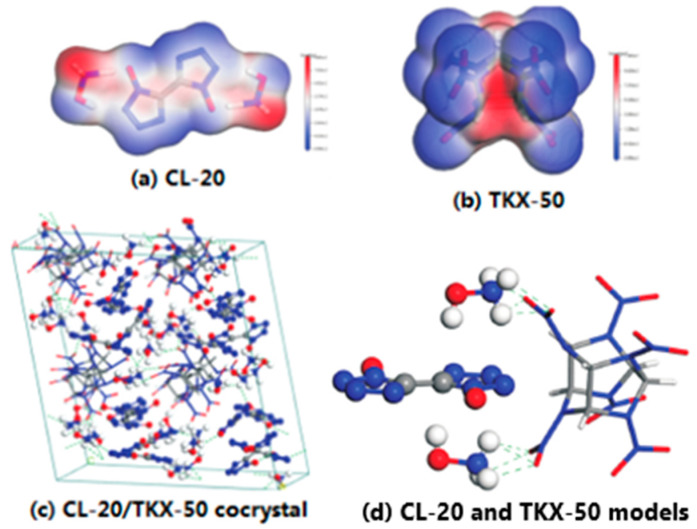
Surface electrostatic potential energy distribution map of CL-20 (**a**) and TKX-50 (**b**), and the equilibrium structure of CL-20/TKX-50 cocrystal model (**c**), CL-20 and TKX-50 models (**d**) [34], with permission from Huo Zha Yao Xue Bao, 2020.

**Figure 4 molecules-25-04311-f004:**
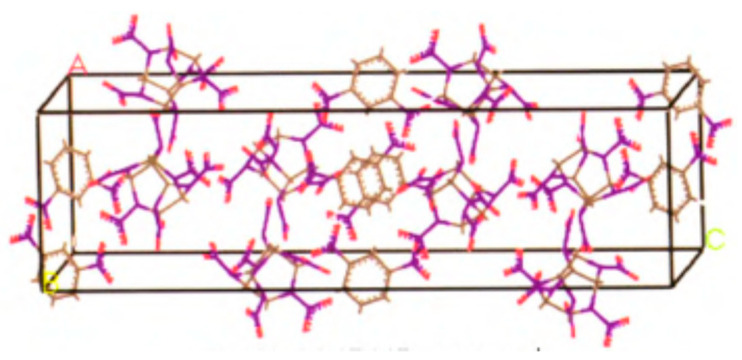
The primitive cell of CL-20/DNB cocrystal [3], with permission from Han Neng Cai Liao, 2015.

**Figure 5 molecules-25-04311-f005:**
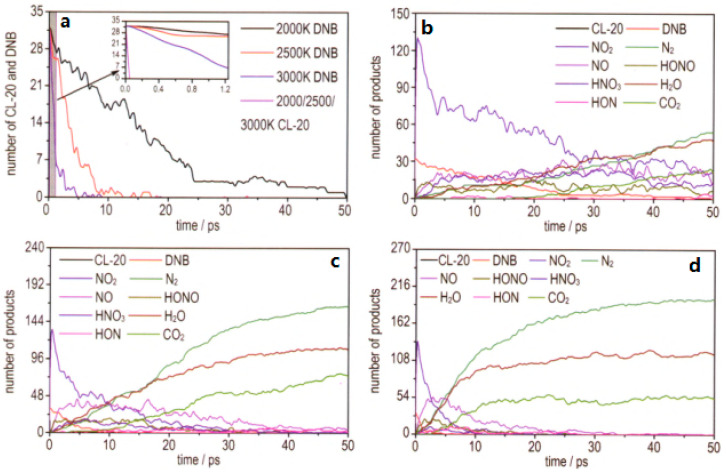
Time evolution of consumption of CL-20 and DNB and main products at various temperatures. (**a**) CL-20 and DNB; (**b**) 2000 K; (**c**) 2500 K; (**d**) 3000 K; [4], with permission from Han Neng Cai Liao, 2016.

**Figure 6 molecules-25-04311-f006:**
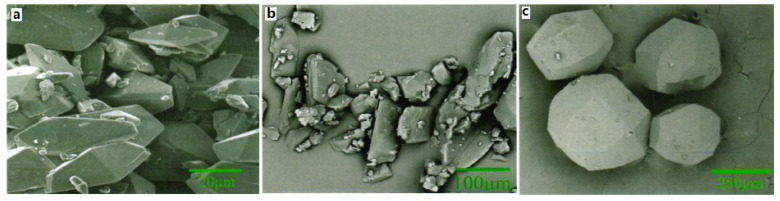
Scanning electron microscope (SEM) photographs of CL-20, TNT and CL-20/TNT cocrystal. (**a**) CL-20; (**b**) TNT; (**c**) CL-20/TNT cocrystal [8], with permission from Han Neng Cai Liao, 2012.

**Figure 7 molecules-25-04311-f007:**
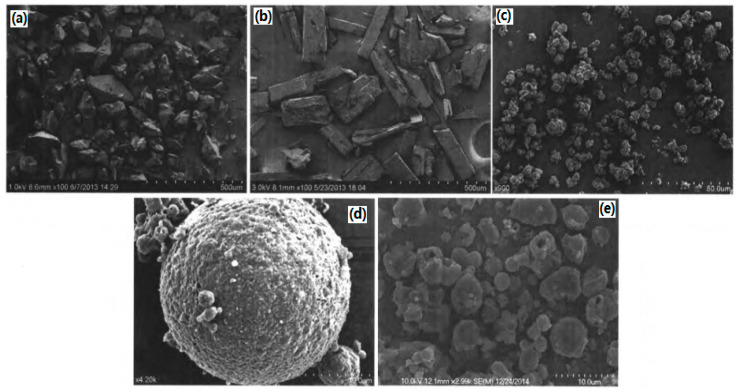
SEM photographs of explosive samples. (**a**) Raw CL-20 (×100); (**b**) Raw TNT (×100); (**c**) CL-20/TNT cocrystal (×900); (**d**) CL-20/TNT cocrystal (×4200); (**e**) Spry drying CL-20 [20], with permission from Han Neng Cai Liao, 2015.

**Figure 8 molecules-25-04311-f008:**
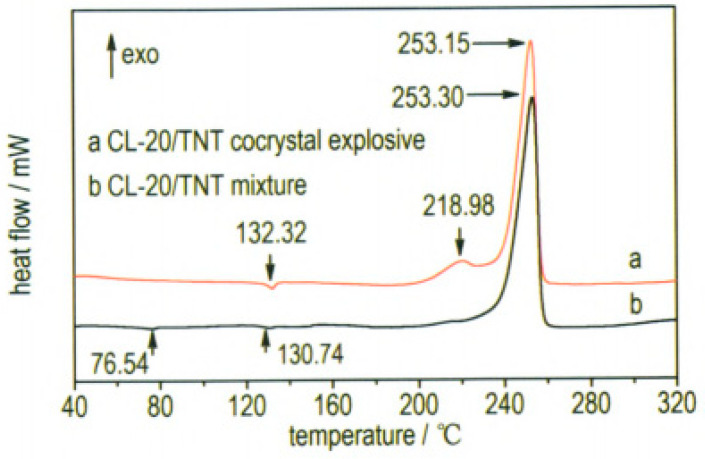
Differential scanning calorimetry (DSC) curves of explosive samples [20], with permission from Han Neng Cai Liao, 2015.

**Figure 9 molecules-25-04311-f009:**
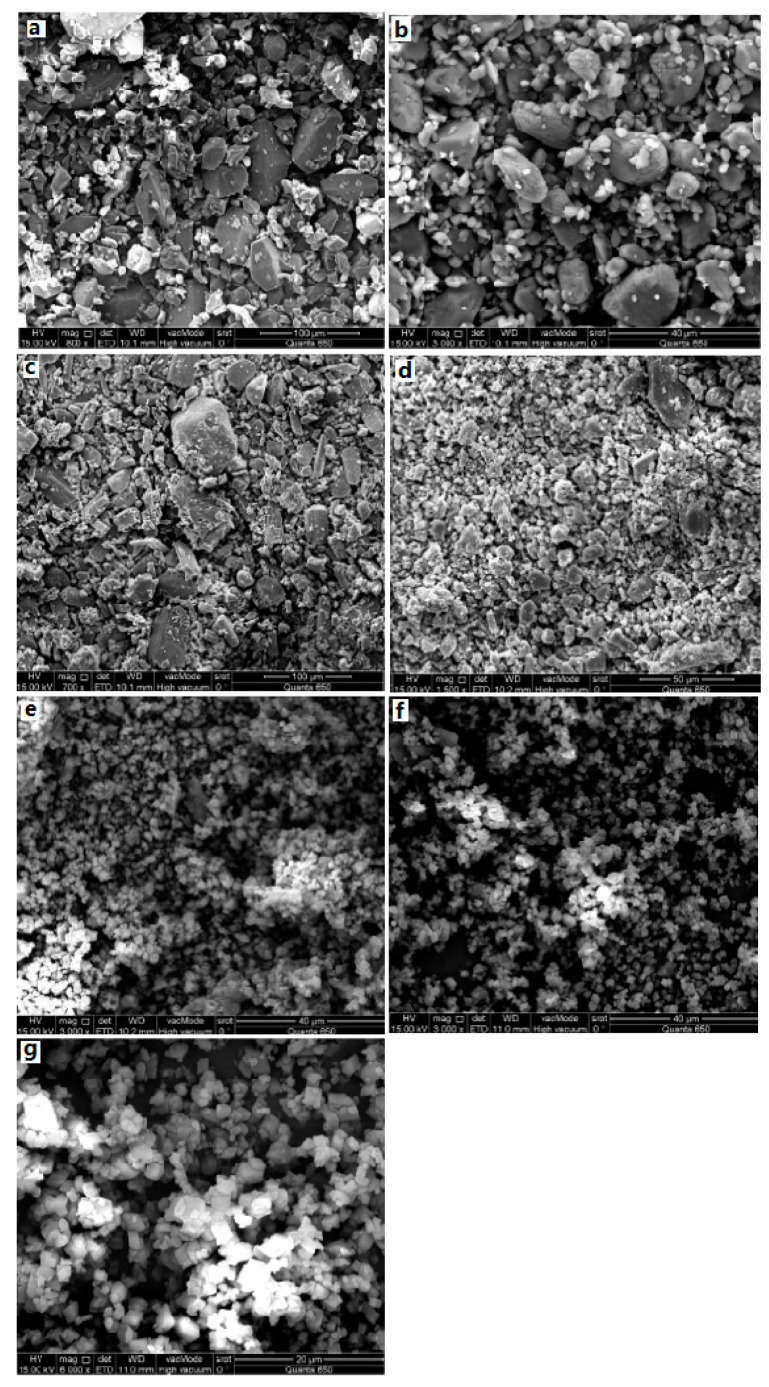
SEM photographs of CL-20, HMX and CL-20/HMX cocrystals at different mixing time. (**a**) CL-20; (**b**) HMX; (**c**) CL-20/HMX cocrystals for 5 min; (**d**) CL-20/HMX cocrystals for 15 min; (**e**) CL-20/HMX cocrystals for 30 min; (**f**) CL-20/HMX cocrystals for 45 min; (**g**) CL-20/HMX cocrystals for 60 min [6], with permission from Han Neng Cai Liao, 2020.

**Figure 10 molecules-25-04311-f010:**
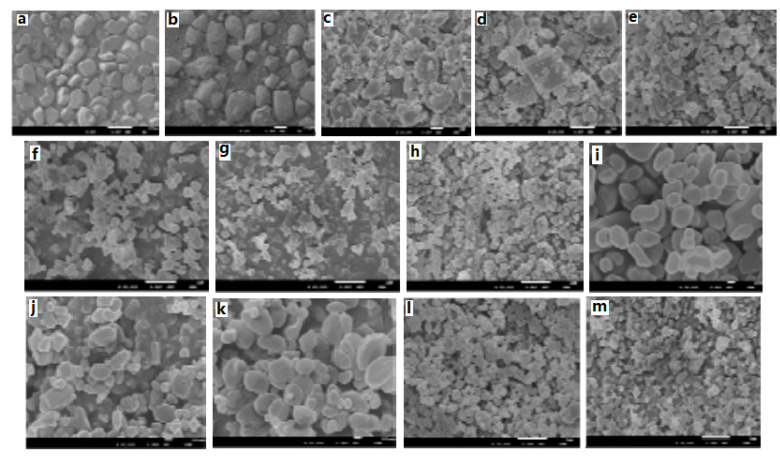
SEM photographs of CL-20/HMX mixture at different milling time. (**a**) 10 min; (**b**) 20 min; (**c**) 30 min; (**d**) 40 min; (**e**) 50 min; (**f**) 60 min; (**g**) 90 min; (**h**) 120 min; (**i**) 180 min; (**j**) 300 min; (**k**) 480 min; (**l**) CL-20; (**m**) HMX [7], with permission from Initiators & Pyrotechnics, 2018.

**Figure 11 molecules-25-04311-f011:**
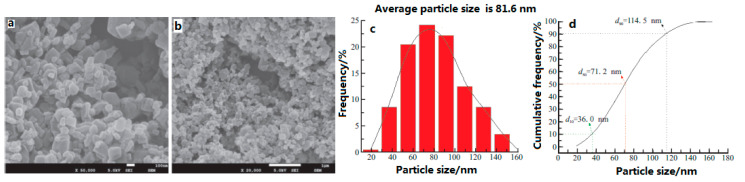
SEM photographs and particle size distribution of nano-sized CL-20/HMX cocrystal. (**a**) 50,000; (**b**) 20,000; (**c**) Frequency distribution; (**d**) Cumulative frequency distribution [9], with permission from Gu Ti Huo Jian Ji Shu, 2018.

**Figure 12 molecules-25-04311-f012:**
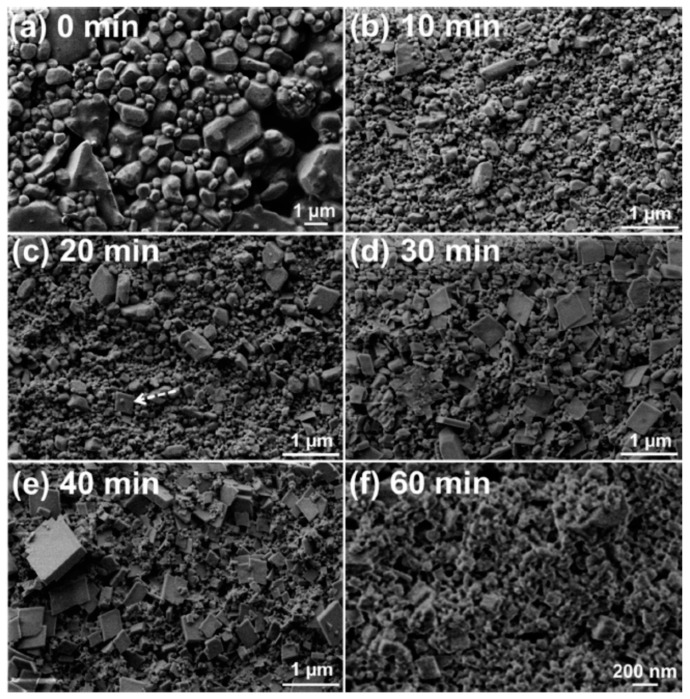
SEM images (**a**–**f**) of specimens sampled at milling times between 0 and 60 min [36], with permission from Cryst. Eng. Comm, 2015.

**Figure 13 molecules-25-04311-f013:**
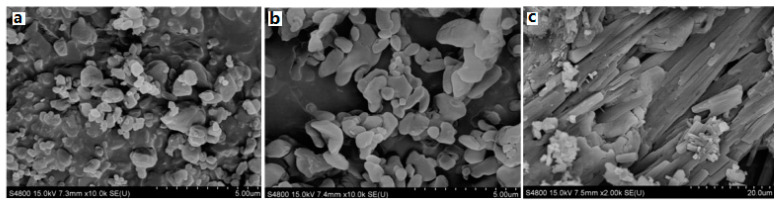
SEM photographs of CL-20, TKX-50 and CL-20/TKX-50 cocrystal. (**a**) raw CL-20; (**b**) Raw TKX-50; (**c**) CL-20/TKX-50 cocrystal [34], with permission from Huo Zha Yao Xue Bao, 2020.

**Figure 14 molecules-25-04311-f014:**
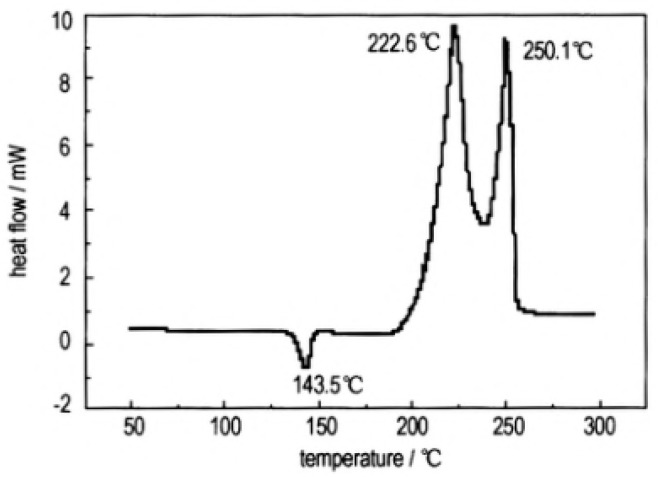
DSC curve of the CL-20/TNT cocrystal explosive [8], with permission from Han Neng Cai Liao, 2012.

**Figure 15 molecules-25-04311-f015:**
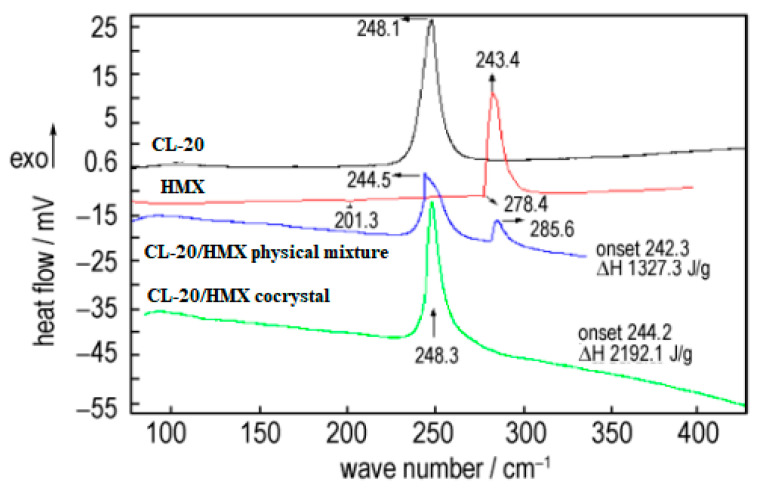
DSC curves of different samples [6], with permission from Han Neng Cai Liao, 2020.

**Figure 16 molecules-25-04311-f016:**
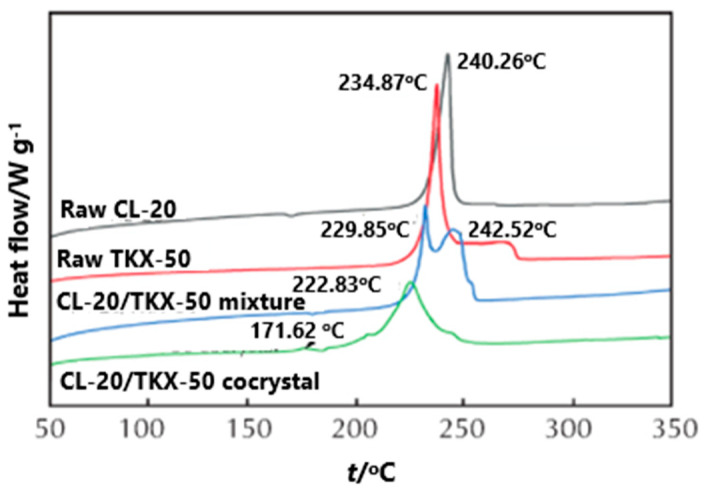
DSC curves of CL-20, TKX-50, CL-20/TKX-50 mixture and CL-20/TKX-50 cocrystal [34], with permission from Huo Zha Yao Xue Bao, 2020.

**Figure 17 molecules-25-04311-f017:**
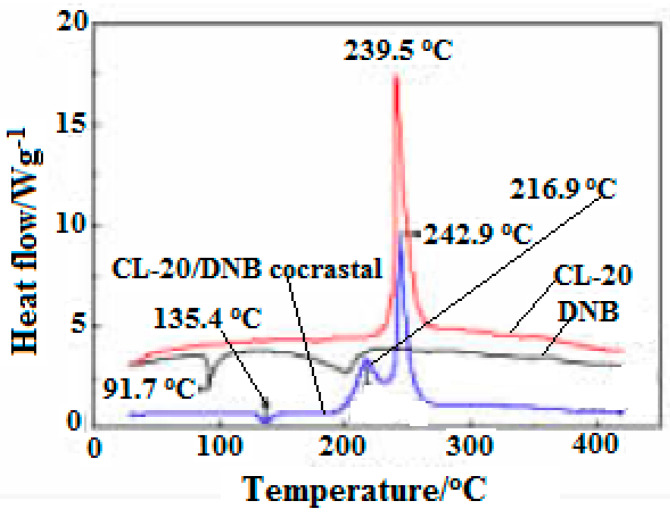
DSC curves of CL-20, DNB and CL-20/DNB cocryatal [5], with permission from Chinese Journal of Energetic Materials, 2013.

**Table 1 molecules-25-04311-t001:** Molecular structures of different energetic materials formed cocrystals.

Samples	Molecular Structures	Samples	Molecular Structures
CL-20	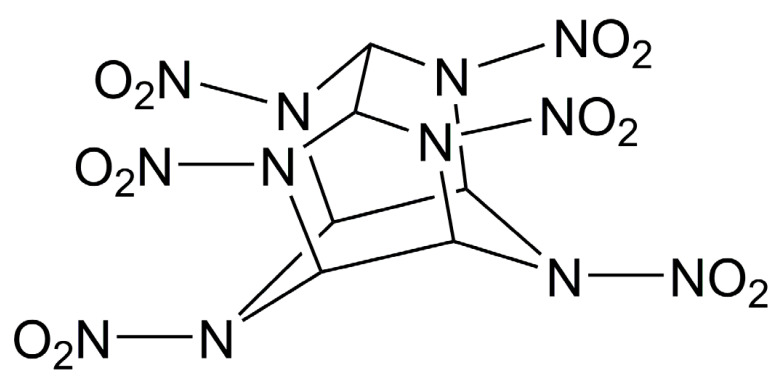	TNT	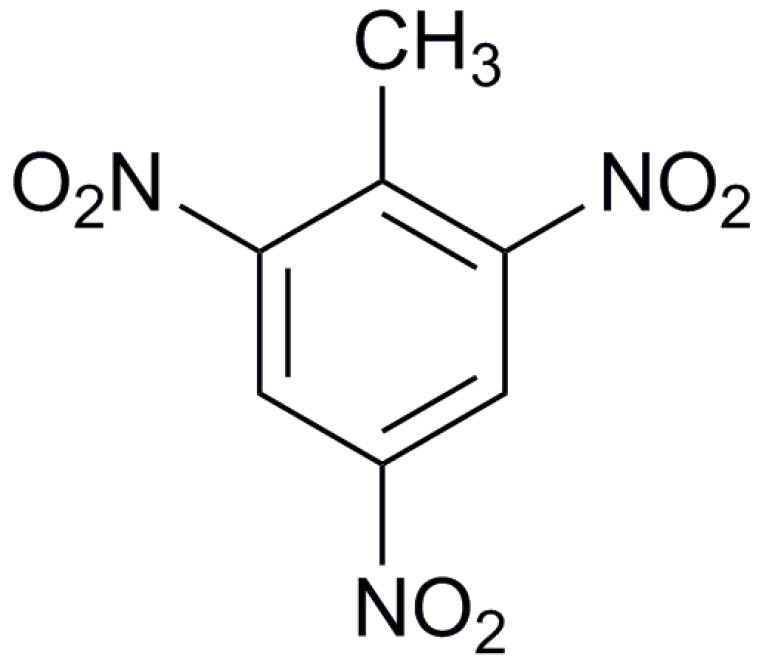
HMX	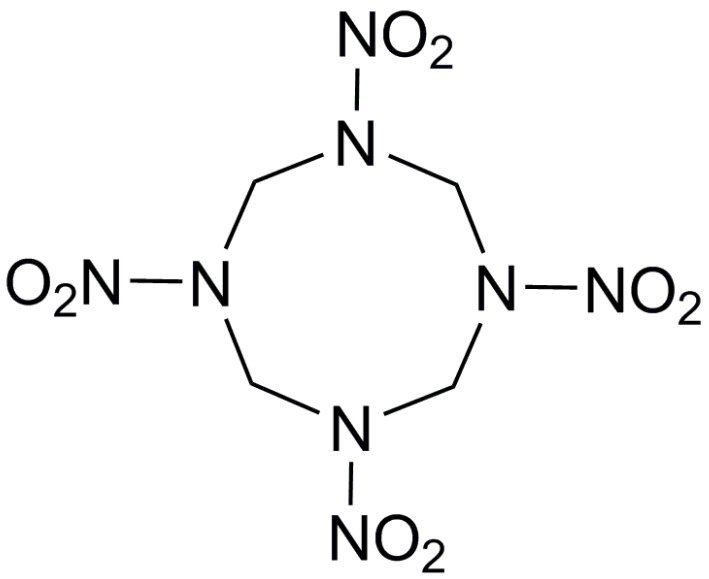	FOX-7	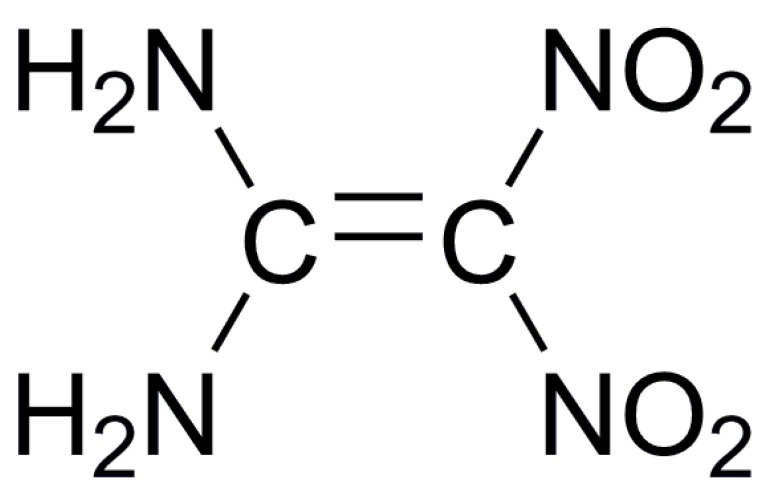
TKX-50	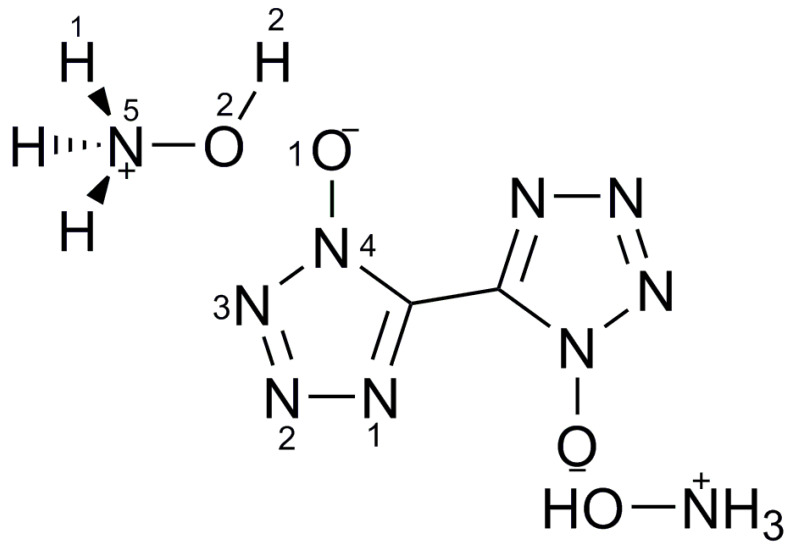	DNB	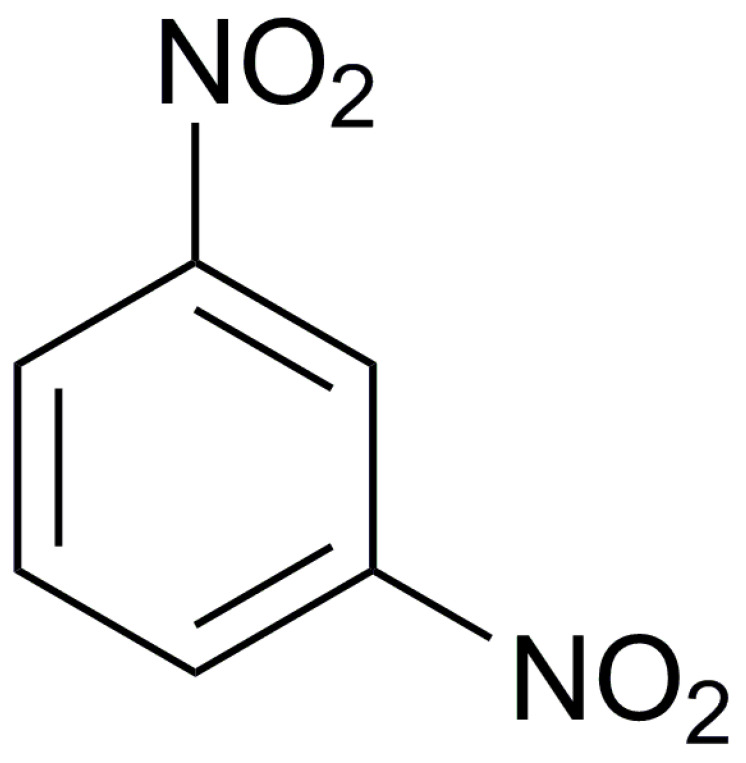

**Table 2 molecules-25-04311-t002:** The cohesive energy density (CED) of CL-20/TNT cocrystal and its CL-20/TNT composite at different temperatures [11], with permission from Han Neng Cai Liao, 2016.

Samples	*T*/K	VdW Energy/kJ·cm^−3^	Electrostatic Energy/kJ·cm^−3^	CED/kJ·cm^−3^
CL-20/TNT cocrystal	195	0.37 (0.01)	0.54 (0.01)	0.91 (0.01)
	245	0.36 (0.00)	0.52 (0.01)	0.88 (0.01)
	295	0.36 (0.00)	0.50 (0.01)	0.86 (0.01)
	345	0.35 (0.00)	0.48 (0.01)	0.83 (0.01)
	395	0.34 (0.00)	0.47 (0.01)	0.81 (0.01)
CL-20/TNT composite	295	0.31 (0.01)	0.45 (0.01)	0.76 (0.01)

Deviations are listed in the parentheses.

**Table 3 molecules-25-04311-t003:** The binding energy between each component of CL-20/HMX mixture, kJ·mol^−1^ [2], with permission from Han Neng Cai Liao, 2016.

Interaction	*E* _total_	*E* _CL-20_	*E* _HMX_	*E* _inter_	*E* _bind_
E	−25,926.37	−6256.29	−395.88	−5718.20	5718.20
vdW	−1656.07	632.68	1187.37	−3363.27	3363.27
Electrostatic	−23,541.63	−7939.87	−15,181.47	−420.30	420.30

*E*_total_ is the single point energy of the equilibrium structure, *E*_CL-20_ is the single point energy of ε-CL-20, *E*_HMX_ is the single point energy of HMX, E is the total energy of each structure, vdW is the energy of each structure obtained by vdW interaction, electrostatic is the energy of each structure obtained by electrostatic interaction, *E*_bind_ is binding energy of CL-20 with HMX, *E*_inter_ is interaction energy of CL-20 with HMX.

**Table 4 molecules-25-04311-t004:** CED of CL-20/HMX cocrystal and mixture systems at different temperatures [2], with permission from Han Neng Cai Liao, 2016.

Samples	Parameters	*T*/K				
		200	250	298	350	400
CL-20/HMX cocrystal	CED	1.167	1.162	1.166	1.151	1.145
	vdW	0.080	0.072	0.074	0.054	0.054
	electrostatic	1.087	1.090	1.092	1.097	1.091
CL-20/HMX mixture	CED	0.069	0.036	0.032	-	0.021
	vdW	0.034				
	electrostatic	0.035	0.038	0.036	0.034	0.036

**Table 5 molecules-25-04311-t005:** The corrected binding energy (in kJ·mol^−1^) of the substituted models of CL-20/FOX-7 [10], with permission from the Journal of Mol Model, 2016.

		8:1	5:1	3:1	2:1	1:1	1:2	1:3	1:5	1:8
ε	(0 1 1)	−337.2	−625.4	−982.3	−1204.2	−1165.2	−1262.8	−1224.7	−844.1	−607.2
	(1 1 0)	−303.0	−578.6	−748.5	−1125.2	−1383.6	−1417.7	−1232.2	−1115.3	−642.2
	(1 0 -1)	−348.2	−619.5	−807.7	−1125.9	-	−1183.1	−1189.3	−806.4	−677.0
	(0 0 2)	−323.2	−523.8	−839.4	−946.4	−1136.5	−1206.7	−787.9	−728.4	−761.1
	(1 1 -1)	−365.2	−490.0	−967.1	−1079.2	−1290.4	−1074.0	−1267.2	−911.7	−624.6
	(0 2 1)	−372.5	−572.9	−964.2	−1031.1	−1368.5				
	(1 0 1)	−416.6	−523.0	−938.2	−1184.7	−1422.7	−1122.6	−927.9	−871.6	−681.2
	Random	−381.5	−581.4	−946.3	−1097.3	−1589.0	−1361.4	−1446.0	−1078.0	−822.1
γ	(0 1 1)	−393.6	−503.1	−918.0	−1023.5	−1231.6	−1288.0	−1140.1	−1028.2	−518.3
	(1 1 0)	−363.5	−511.2	−940.2	−1101.6	−1456.7	−1605.1	−1387.0	−1179.4	−735.1
	(1 0 -1)	−452.6	−543.1	−815.2	−1128.4	−1489.2	−1588.2	−1311.8	−1056.6	−726.0
	(0 0 2)	−428.3	−511.6	−762.3	−878.5	−1003.9	−1257.3	−1157.3	−967.8	−732.3
	(1 1 -1)	−420.3	−493.1	−659.3	−815.2	−979.2	−1137.0	−1024.5	−922.3	−657.1
	(0 2 1)	−466.0	−526.3	−762.1	−912.1	−1218.1				
	(1 0 1)	−431.8	−527.6	−988.6	−1215.1	−1345.5	−1322.1	−1252.3	−977.8	−511.6
	Random	−378.9	−515.8	−927.5	−1288.4	−1587.2	−1369.7	−1201.5	−835.6	−417.5
β	(0 0 1)	−367.0	−578.9	−913.4	−1007.5	−1137.6				
	(0 1 0)	−389.1	−618.0	−985.6	−1123.7	−1165.2				
	Random	−408.5	−579.3	−866.2	−1252.8	−1443.6	−1532.7	−1276.8	−828.1	−497.6

**Table 6 molecules-25-04311-t006:** Total energies, component energies, binding energy and its normalized values of two different polymerbonded explosives (PBXs) [3], with permission from Han Neng Cai Liao, 2015.

Samples	*E* _total_	*E* _base_	*E* _polymer_	*E* _bind_	*E*’_bind_
CL-20/DNB/HTPB	−51,789.1	−51,322.6	343.0	809.4	185.2
	(114.1)	(108.0)	(29.5)	(31.3)	(7.2)
CL-20/DNB/PEG	−51,451.5	−51,367.3	858.4	942.6	214.2
	(134.5)	(124.9)	(42.1)	(42.6)	(9.7)

**Table 7 molecules-25-04311-t007:** The lattice parameters of CL-20/DNB cocrystal were calculated by the ReaxFF force field compared with the experimental value.

Lattice Parameter	Experimental Value [5]	ReaxFF	Error/%
*a*/Å	9.4703	9.5606	0.95
*b*/Å	13.4589	13.5872	0.95
*c*/Å	33.620	33.9406	0.95
Density/g·cm^−3^	1.880	1.8268	−2.8

**Table 8 molecules-25-04311-t008:** The mechanical sensitivities of different samples.

Samples	*F*/%	*H*_50_/cm	*P*/MPa	*D*/m·s^−1^
CL-20	100 [6,7]	19.3, 13.1 [20]; 13.1 [7]; 15 [8,9,34]	44.9 [34]; 43 [8]	9385 [34]; 9500 [8]
HMX	28 [6]; 84 [7]	36.4 [6]; 19.6 [7]; 27.2 [9]; 13.9 [34]	39.6 [34]	9048 [34]
Ball milling of CL-20	72 [7]	42.3 [7]		
Ball milling of HMX	60 [7]	47.8 [7]		
CL-20/HMX mixture	96 (mole ratio is 2:1) [7]	15.4 (mole ratio is 2:1) [7], 20.1 [9]		
Ball milling of CL-20/HMX mixture (mole ratio is 2:1)	68 [7]	43.5 [7]		
Micro/nano CL-20/HMX cocrystal	60 [7]	47.6 [7], 32.6 [9]		
Ultrafine CL-20/HMX cocrystal	84 [6]	47.9 [6]		
TNT		102 [8], 157.2 [20]	21 [8]	6900 [8]
CL-20/TNT cocrystal		28 [8], 49.3 [20]	35 [8]	8600 [8]
CL-20/TNT mixture		18.8 [20]		
TKX-50		55.4 [34]	41.0 [34]	8524 [34]
CL-20/TKX-50 mixture		26.0 [34]		
CL-20/TKX-50 cocrystal		34.0 [34]	43.8 [34]	9264 [34]

*H*_50_ is characteristic drop height, cm; *F* is friction sensitivity, %; *D* is detonation velocity, m·s^−1^; *P* is detonation pressure, MPa. The hammer weight: (2.500 ± 0.002) kg, sample weight: (35 ± 1) mg for impact sensitivity in [7,20], the hammer weight: (1.5 ± 0.01) kg, sample weight: (20 ± 1) mg, pressure: (2.45 ± 0.07) MPa, swing angle: (80 ± 1)^0^ for friction sensitivity in [7]; the hammer weight: 2 kg, sample weight: 30 mg for impact sensitivity, for friction sensitivity in [6]; the hammer weight: 2 kg, sample weight: (30 ± 1) mg in [8]; the hammer weight: 5 kg, sample weight: (35 ± 1) mg in [9,34].

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
