# Peer review of "CL-20-Based Cocrystal Energetic Materials: Simulation, Preparation and Performance"

_molecules, 2020, doi:10.3390/molecules25184311_

Round 1

Reviewer 1 Report

Line 16. Spare dot: stimuli. and cocrystallization.

Table 1. Structures are different quality, please insert structures with the same high quality.

Line 106. (2.04 g/cm3, 3210 oC) - three thousands degrees?

Table 8. The data are showed unclear. Please add a additional collumn for VOD and PCJ. Please add a unit/s.

Author Response

Comments and Suggestions for Authors 1

Line 16. Spare dot: stimuli. and cocrystallization.

Reply: The dot was changed to comma.

Table 1. Structures are different quality, please insert structures with the same high quality.

Reply: The structures of materials were repainted.

Line 106. (2.04 g/cm3, 3210 oC) - three thousands degrees?

Reply: It should be 210 oC, it has been revised in the text.

Table 8. The data are showed unclear. Please add a additional collumn for VOD and PCJ. Please add a unit/s.

Reply: Addition columns of VOD and PCJ were provided, and the Table was improved.

Reviewer 2 Report

The grammar and proofing is not very good. I am not going to copy edit but start with the abstract. There is a period after ‘stimuli.’ While it could be a slight mistake, no matter how you take the section after (either as a new sentence or continuation of the previous one) it does not make sense.

Mistakes in references (look at number 20 for example). Some of their refetrences are odd. Number 21 highlights how co-crystals are a ‘hot’ area. Come on.

Please increase the resolution of the images/figures (this may be an internal problem when constructing the PDF)

The modelling is what it is. Using Accelrys force fields to minimize the energy and the thermocode by Goddard is OK. Here is the problem:

There is no evidence that they have actually made the cocrystals experimentally. While the TNT system has been published the authors are relying on DSC. That is a very inappropriate approach as colligative properties alone can shift peaks.   At least provide xrd to prove you have a co-crystal. With the x-ray structure the authors can then have experimental structures to compare in their thermochemical code.

I also worry about any potential impact of this work since some of it has been published in more detail including the comments I have above) J. Chem. Inf. Model. 2019, 59, 5, 2079–2092

https://pubs.acs.org/doi/pdfplus/10.1021/acs.jcim.8b00952?src=recsys

DOI: 10.30919/esmm5f126

While it is not inappropriate, it is a shame that they used so many references that I cannot compare data when there are multiple references of similar work out there.

Author Response

Comments and Suggestions for Authors 2

The grammar and proofing is not very good. I am not going to copy edit but start with the abstract. There is a period after ‘stimuli.’ While it could be a slight mistake, no matter how you take the section after (either as a new sentence or continuation of the previous one) it does not make sense.

Reply: The grammar and proofing were modified and improved. The period was deleted.

Mistakes in references (look at number 20 for example). Some of their references are odd. Number 21 highlights how co-crystals are a ‘hot’ area. Come on.

Reply: The CL-20/TNT cocrystal was investigated in reference 20, and it is much close to our manuscript topic. I don’t know what is the meaning of odd. Is it “odd” or “old”?

Please increase the resolution of the images/figures (this may be an internal problem when constructing the PDF)

Reply: The resolution of Figures were improved.

The modelling is what it is. Using Accelrys force fields to minimize the energy and the thermocode by Goddard is OK. Here is the problem:

There is no evidence that they have actually made the cocrystals experimentally. While the TNT system has been published the authors are relying on DSC. That is a very inappropriate approach as colligative properties alone can shift peaks.  At least provide xrd to prove you have a co-crystal. With the x-ray structure the authors can then have experimental structures to compare in their thermochemical code.

Reply: You are right, x-ray it much important on the characterization of the sample structures.

The XRD pattens were provided in the original papers, we can forward them according to the citation in the text, here please allow me focus on the performance of cocrystals based on the prepared cocrystal samples.

I also worry about any potential impact of this work since some of it has been published in more detail including the comments I have above) J. Chem. Inf. Model. 2019, 59, 5, 2079–2092

https://pubs.acs.org/doi/pdfplus/10.1021/acs.jcim.8b00952?src=recsys

DOI: 10.30919/esmm5f126

While it is not inappropriate, it is a shame that they used so many references that I cannot compare data when there are multiple references of similar work out there.

Reply: There are only papers either on simulation or experiments on cocrystal, this is one overview paper on simulation, preparation and performance of CL-20-based cocrystals with other typical compounds.

For the multiple references of similar work, we can compare the prepared and characterized methods on cocrystals, including particle size, size distribution, and measuring parameters, etc.
